# Metabolic Responses to Manganese Toxicity in Soybean Roots and Leaves

**DOI:** 10.3390/plants12203615

**Published:** 2023-10-19

**Authors:** Yanyan Wang, Jianyu Li, Yuhu Pan, Jingye Chen, Ying Liu

**Affiliations:** Department of Biotechnology, College of Coastal Agricultural Sciences, Guangdong Ocean University, Zhanjiang 524088, China; yanyanwang@gdou.edu.cn (Y.W.);

**Keywords:** *Glycine max*, Mn toxicity, metabolomics, roots, leaves

## Abstract

Soybean is one of the most crucial beans in the world. Although Mn (manganese) is a kind of important nutritive element helpful to plant growth and health, excess Mn is harmful to crops. Nevertheless, the effect of Mn toxicity on soybean roots and leaves metabolism is still not clear. To explore this, water culture experiments were conducted on the development, activity of enzyme, and metabolic process of soybeans under varying levels of Mn treatment (5 and 100 μM). Compared with the control, the soybeans under Mn stress showed inhibited growth and development. Moreover, the activity of superoxide dismutase (SOD), catalase (CAT), peroxidase (POD), ascorbate peroxidase (APX), and the soluble protein content in leaves and roots of soybean were all increased. However, soluble sugar and proline contents in soybean roots and leaves showed the opposite trend. In addition, the Mg (magnesium) and Fe (iron) ion contents in soybean leaves significantly decreased, and the Mn ion content greatly increased. In roots, the Mn and Fe ion content increased, whereas the Mg ion content decreased. Furthermore, the metabolomic analysis based on nontargeted liquid chromatography–mass spectrometry identified 136 and 164 differential metabolites (DMs) that responded to Mn toxicity in roots and leaves of soybean, respectively. These DMs might participate in five different primary metabolic pathways in soybean leaves and roots, suggesting that soybean leaves and roots demonstrate different kinds of reactions in response to Mn toxicity. These findings indicate that Mn toxicity will result in enzymes activity being changed and the metabolic pathway being seriously affected, hence inhibiting the development of soybean.

## 1. Introduction

Mn (manganese) is one kind of trace element essential to plants [1]. Mn participates in various plant metabolic processes, such as oxygenic photosynthesis, respiratory action, synthesis of proteins and fatty acids, and activation of enzymes [2]. For instance, the role of Mn in plants as a cofactor and a catalyst of electron transport in photosynthesis is crucial [3]. Mn deficiency reduces photosynthetic electron transport and oxidative stress, adversely affecting the photosynthetic organs [4,5]. Mn is also a helper factor of numerous enzymes, decarboxylases in the TCA (tricarboxylic acid) cycle, RNA (ribonucleic acid) polymerases, SOD (superoxide dismutase), and suchlike [1]. Moreover, Mn is involved in the secondary metabolite synthesis, such as flavonoids, lignins, and suchlike [6].

Optimal Mn nutrition contributes to plant development, but excess Mn can be harmful to many plants [3]. For most plants, approximately 20–40 mg per kg (dry weight) is enough for normal development [7]. The contents of Mn in leaves of various kinds of plants are different, generally 30–500 mg per kg (dry weight) [8]. However, Mn is extremely toxic to plants when its concentration exceeds the threshold in crops. Previous studies demonstrated that Mn stress can inhibit the absorption and transport of other elements such as iron (Fe), magnesium (Mg), calcium (Ca), and phosphorus (P), disrupt the structure of chloroplast, and impede photosynthesis and respiration [9,10,11]. As a result, plants exhibit visible poisoning symptoms on plant leaves, including chlorosis, brown spots, and crinkled leaves [12,13]. In addition to leaves, roots are affected by Mn toxicity, such as decreased number of lateral roots and dry weight [14,15].

Plants produce various physiological changes and biochemical responses to relieve the poisoning symptoms of metal ion excess [16]. The most common coping strategy is activating the antioxidant system’s related enzymes, containing ascorbate peroxidase (APX), catalase (CAT), peroxidase (POD), and SOD [17,18,19]. For example, studies have shown that the Mn resistant ryegrass (*Lolium perenne*) cultivar named ‘Kingston’ displays more activity of SOD than the Mn sensitivity cultivar named ‘Nui’ [18]. On the contrary, plants improve Mn tolerance by regulating Mn uptake, translocation, and distribution [1]. The Nramp (natural resistance-associated macrophage protein) family is one of the major Mn transporters [19]. Numerous Mn transporters, such as *AtNramp1* and *AtNramp3* from *Arabidopsis* and *OsNramp1* and *OsNramp5*, have been identified from rice (*Oryza sativa*) [19,20]. In addition, plant roots secrete organic acids to chelate Mn^2+^ in the soil, thereby forming stable metal chelates for reducing Mn toxicity [21].

Metabolites can perform numerous functions, and the quantitative and qualitative analyses of them can authenticate the state of metabolic response in plants with biotic and abiotic stresses [22]. Thus, the development of omics technology has resulted in the wide use of metabolomics for detecting the changes in plant metabolites under environmental stresses. Zhang et al. conducted metabolomics analysis to reveal how cucumbers (*Cucumis sativus*) reprogram metabolic products to handle silver (Ag^+^) and silver nanoparticles (AgNPs) that induce oxidative stress (2018). The results showed that AgNPs enhance respiration, inhibit photorespiration, and induce production of *p*-benzoquinone, lactulose, carbazole, citraconic acid, acetanilide, lactamide, raffinose, and so on [23]. Wang et al. (2023) reported that flavonoids may play an important part in Mn tolerance; however, the mechanism was unclear for flavonoids regulating Mn tolerance [24]. Moreover, ATP-binding cassette (ABC) transporters and amino acid biosynthesis were enriched under Mn stress [24].

Soybean (*Glycine max* L.) is an important oil crop and a source of animal feed [25,26]. Mn stress is an important factor restricting soybean yield and quality. Previous research showed that soybean biomass decreases at 200 μM Mn [10]. Researchers have recently conducted omics analysis to investigate the mechanisms of soybean responses to Mn toxicity. Some research results demonstrated that the growth of soybean root is markedly restrained by Mn poisoning; they also identified protein synthesis metabolism, the component of cell walls, and signaling pathway via proteomic analyses [10]. Transcriptome analyses imply that the genes, coding ABC transporter, LEA (late embryogenesis abundant) protein, ion transporter, and aldehyde dehydrogenase respond to Mn stress [27]. However, metabolomics analyses of soybean in response to Mn poisoning are very few, while metabolome analyses of soybean leaves and roots exposed to manganese toxicity have not been reported. In current study, the phenotypic changes and physiological response indexes of soybean leaves and roots after high and normal Mn treatment were analyzed. Then, in order to further study the specific metabolic regulation mechanism of soybean subjected to manganese toxicity stress, the changes in metabolites with normal and high concentrations of Mn treatment were analyzed through metabolomics analysis. This study provides useful information for further revealing the Mn tolerance mechanism in soybean, and lays a foundation for genetic improvement of Mn tolerance traits in crops.

## 2. Results

### 2.1. Influence of Mn Stresses on the Growth of Soybean 

The result showed that Mn poisoning played only a minor role in the height of soybean seedlings (Figure 1A,B). However, it seriously reduced the stem width and the raw weight of shoot and root tissues (Figure 1D–F). Compared with the 5 µM Mn control group, the 100 µM Mn treatment group reduced the stem width by 0.37 mm (Figure 1C). In addition, the raw weight of shoot and root decreased by 18.7% and 35.4%, respectively (Figure 1E,F).

### 2.2. Effects of Mn Toxicity Stress on Roots and Leaves of Soybean

The influence of Mn poisoning on the development of soybean roots and leaves increased with the increase in exogenous Mn concentration. The leaves exhibited poisonous spots, but little effect was found on the chlorophyll content. The spot density under the 100 μM Mn treatment increased by 31.8 times compared with that under 5 μM Mn concentration (Figure 2A,B; Table 1). Compared with those of the control, root diameter, root volume, root surface area, and total root length exposed to 100 µM Mn concentration remarkably decreased by 8.1%, 37.1%, 37.1%, and 33.1%, respectively; however, the root tip number increased by 21.4% (Figure 2C,D; Table 1).

### 2.3. Change in Mn, Fe, and Mg Contents in Roots and Leaves of Soybean Suffering Mn Toxicity Stress

The Fe, Mg, and Mn concentrations in roots and leaves of soybean were surveyed at different Mn treatment levels (Figure 3). The Mn content increased in leaves and roots of soybean suffering Mn poisoning. The contents of Mn in roots and leaves at 100 µM Mn concentration were 4.6 and 4.9 times those at 5 µM Mn concentration (Figure 3A). The contents of Mg in the roots and leaves of soybean significantly decreased when the soybean seedlings were treated with 100 µM Mn; they reduced by 37.3% and 8.9% when the soybean seedlings were treated with 5 µM (Figure 3C). The trends of Fe accumulation in roots and leaves varied under two Mn concentrations. Compared with the 5 µM Mn treatment group, the 100 µM Mn experimental group showed a decreased Fe content in the leaves by 78.8% and an increased Fe content in the roots by 368.0% (Figure 3B). 

### 2.4. Change in the Physiological Response Indicators in Roots and Leaves of Soybean at Different Mn Treatment

Many physiological response indicators were affected by soybean Mn poisoning, and the responses of many physiological parameters in soybean leaves and roots to Mn poisoning were different (Figure 4). Compared with the control, the soybean roots under Mn stress exhibited increased POD, CAT, APX, and SOD activities by 180.9%, 225.0%, 61.3%, and 49.3%, respectively. 

The corresponding enzymes increased by 222.5%, 64.2%, 43.2%, and 83.0% in leaves. When the Mn concentration increased from 5 µM to 100 µM, the soluble protein and malondialdehyde (MDA) contents in soybean roots increased by 46.99% and 50%, respectively, whereas the proline content decreased by 36.6%; the soluble sugar did not show any significant changes. For soybean leaves, the soluble protein and soluble sugar increased by 21.8% and 22.8%, respectively, whereas the MDA and proline contents in the leaves showed no significant changes.

### 2.5. Metabolic Profiling of Soybean Leaves and Roots in Responding to Mn Poisoning

We detected the metabolism in soybean roots and leaves under 5 and 100 µM Mn treatment to understand how soybean leaves and roots respond to Mn poisoning. The total metabolites of root and leaves were acquired and tested via broad-spectrum nontargeted metabolome analysis with the UPLC–MS/MS (Thermo Fisher Scientific, Waltham, MA, USA) (ultra performance liquid chromatography–tandem mass spectrometry) method. All of the 6876 metabolic products were characterized in the roots and leaves of the soybean. First, PCA (principal component analysis) was conducted to obtain credible results (Figure 5A). All of the samples’ PCA demonstrated that there was little difference within the groups, whereas great variations were observed between the groups (Figure 5A). The two main constituents of PCA accounted for 63.2% of the total variance. The first principal component accounted for 54.7% of the total variation. Therefore, the metabolisms of roots and leaves were separate. The metabolisms of the Mn treatment and control groups were also separate. Second, the OPLS-DA (orthogonal partial least squares discriminant analysis), a method of supervised clustering, usually offers stronger analytical skills than PCA [28]. The OPLS-DA multivariate analysis based on the GC–MS (gas chromatography–mass spectrometry) dataset was performed to maximize the separation between the treatment groups (Figure 5B). Similarly, the metabolic profiles differed among the RCK (roots-CK, 5 µM Mn treatment), R100 (roots, 100 µM Mn treatment), LCK (leaves-CK, 5 µM Mn treatment), and L100 (leaves, 100 µM Mn treatment) groups. These results indicated that the Mn poisoning markedly influenced the metabolic of roots and leaves, revealing that the method adopted in this study properly displayed the basal metabolic and different effects of Mn poisoning on soybean roots and leaves.

### 2.6. Differential Metabolites (DMs) in Soybean Roots and Leaves in Response to Mn Poisoning

The values of log2 FC (fold change) and VIP (variable importance in projection) of the OPLS-DA models were joined together with screen DMs and we identified the responses of DMs in roots and leaves to Mn poisoning. Then, 136 (upregulated, 46; downregulated, 90) and 164 (upregulated, 101; downregulated, 63) metabolites were identified separately in roots and leaves between the treatment and control groups (Figure 6A,B; Appendix A). The above 300 metabolic products were selected on the basis of the following condition: (1) metabolic products with absolute log2 FC > 1 were chosen; (2) on the basis of the above test results, metabolic products with VIP > 1 and *p* < 0.05 were chosen. In total, 55 DMs were found in both leaves and roots of soybean (Figure 6C). In addition, 109 DMs responded particularly to Mn toxicity in leaves, and 81 distinctive metabolites were produced specifically in roots (Figure 6A,C). These results suggest that leaves and roots showed identical and specific responses to Mn toxicity.

### 2.7. Comparative Analysis of Metabolic Pathways Responding to Mn Toxicity in Leaves and Roots

DMs were divided into 15 classes by chemical properties in roots and leaves (Appendix A) to investigate the metabolic responses of leaves and roots in soybean under Mn treatment. Fatty acyls, organic compounds, prenols, carboxylic acids, flavonoids, nucleosides, and benzene were significantly affected by Mn toxicity in roots and leaves. However, this finding does not prove that the metabolic response is the same between roots and leaves under Mn stress. Thus, KEGG analysis was implemented for DMs in soybean leaves and roots (Appendix A). The findings displayed that the DMs of leaves and roots were primarily mapped to 20 different plant metabolic pathways, respectively. Furthermore, the important pathways of Mn poisoning were selected by searching the MetPA (Metabolomics Pathway Analysis) database to mine the key metabolic pathway. The five metabolic pathways, (1) metabolism of linoleic acid, (2) metabolism of purine, (3) glycolysis/gluconeogenesis, (4) alpha-linoleic acid metabolism, and (5) pyrimidine metabolism, were identified to play pivotal roles in the root response to Mn toxicity (Figure 7A). Five key pathways were identified in the leaves: (1) flavonoid biosynthesis, (2) citrate cycle (TCA cycle), (3) oxidative phosphorylation, (4) ABC transporters, and (5) metabolisms of aspartate, glutamate, and alanine (Figure 7B). On the basis of the above findings, we proposed a simplified model of how to respond to the Mn toxicity of soybean roots and leaves at the metabolic level (Figure 8). Therefore, the comparison of the pathways that were markedly influenced by roots and leaves indicated that Mn toxicity on soybean roots and leaves mainly affected the metabolic activity of plants by the different metabolism pathways.

## 3. Discussion

Mn is one of the vital minor elements for plant development, and it is also involved in maintaining metabolic activity in various plant cells [29]. Nevertheless, the stable state of Mn will be broken when suffering excess Mn treatment conditions, leading to Mn toxicity. Different crops of species have various responses and tolerances to Mn toxicity [10,30,31]. For example, the dry weights of roots and shoots are decreased in ryegrass (*Lolium perenne*) under 150 µM Mn treatment [32]. In canola (*Brassica napus*), necrotic leaf spots and chlorosis in the leaf margin occur at 200 µM Mn concentration [33]. Dark brown necrotic spots occur on the leaves of barley (*Hordeum vulgare* L.) when the Mn concentration is 1830 µM [34]. In the current study, the fresh weights of the aboveground and underground tissues reduced markedly when the Mn treatment concentration enhanced from 5 µM to 100 µM (Figure 1E,F). In addition, the spot density and root-related indicators decreased remarkably (Figure 2A,B; Table 1). These results suggest that excessive Mn inhibits soybean growth and development, and the soybean is apparently sensitive to Mn stress. 

Thus far, researchers have focused on the response mechanism of plants to heavy metal toxicity via omics analysis [35,36,37]. For example, Duan et al. (2023) studied the influences of antimony (Sb) stress on the root metabolic activity of Nipponbare rice (*Oryza sativa*) [35]. They found that Sb stress affects the TCA cycle, alpha-linolenic acid metabolism, alanine metabolism, and aspartate metabolism [35]. Xu et al. (2016) showed that the tolerance of two tea (*Camellia sinensis*) varieties to aluminum toxicity has a correlation with the DMs of sugars, shikimic acids, and amino acids [38]. For Mn toxicity, previous studies mainly showed the different responses on a molecular level in crops under Mn toxicity via transcriptome analysis [14,27]. However, the level and types of metabolites remained unclear in various tissues of crops under Mn toxicity. This finding was verified by a comparison analysis of roots and leaves of soybean suffering Mn poisoning. In the present study, the responses of DMs to Mn toxicity stress were identified on soybean roots and leaves via metabolomic analysis (i.e., 136 and 164 DMs, respectively). They were screened from soybean roots and leaves (Figure 6). In total, 190 DMs significantly differed between roots and leaves. The result implies that the molecular regulating mechanisms were highly different in soybean leaves and roots suffering Mn poisoning. Furthermore, DMs in soybean leaves and roots were identified to involve various metabolic pathways via KEGG analysis (Figure 7). The top five metabolic pathways were significantly different in soybean leaves and roots (Figure 8). This finding indicates that soybean leaves and roots might have different response mechanisms to Mn toxicity.

In our study, flavonoid biosynthesis, citrate cycle, oxidative phosphorylation, ABC transporters, and metabolisms of aspartate, glutamate, and alanine were significantly enriched in soybean leaves under 100 µM Mn treatment (Figure 7B). Flavonoids perform various functions, such as conferring a wide color range to plants, regulating plant growth and development, and resisting various stresses [39]. For example, *Lupinus luteus* with lead (Pb) treatment increases the total flavonoid contents by 67% and 54% in cotyledons and roots [40]. Furthermore, seedling roots of *L. luteus* preprocessing with total flavone extracted from Pb-treated cotyledons exhibit strengthened toleration to heavy metal stress [40]. Luteolin, a common flavonoid, has an inhibitory effect on the toxicity induced by heavy metals [41]. For example, luteolin regulates the redox imbalance, preserves mitochondrial function, and depresses caspase-family-associated apoptosis induced by copper (Cu) [42]. Therefore, flavonoids may be an effective antidote for preventing the combined toxicity induced by Mn toxicity. 

The citrate cycle is an energy supply process that plays a vital part in the metabolism and energy supply of organisms [43]. Mn treatment leads to changes in the citrate cycle of soybean leaves; these changes are similar to those in other plants treated with metal(loid)s [35,44,45]. In this study, oxaloacetic acid, citric acid, fumaric acid, and malic acid are critical intermediate products in the citric acid cycle; these intermediates have high accumulation (Figure 8). The increase in these intermediates indicates the enhancement of the citrate cycle, which may lead to enough energy for soybean to repair biomolecules damaged by Mn stress. ATP binding cassette (ABC) transporters are primary transporters responsible for energy transport [46]. Some studies suggested that ABC transporters can participate in plant metal transporting [47,48,49]. For example, the pathway of the ABC transporter is markedly enhanced in *Celosia argentea* leaves suffering cadmium (Cd) and Mn poisoning [50]. In the present study, similar results were obtained in soybean leaves under Mn toxicity (Figure 8). This finding indicates the important status of the ABC transporter tolerance to Mn in plants. 

Compared with soybean leaves, the soybean roots under 100 Mn µM treatment exhibited enriched metabolisms of linoleic acid, purine, glycolysis/gluconeogenesis, alpha-linoleic acid, and pyrimidine (Figure 7A and Figure 8). In previous studies, the researchers found that the concentrations of uridine monophosphate, guanosine monophosphate, xanthine, adenosine, guanosine, thymidine, and cytosine monophosphate are downregulated in rice with Cu treatment [51,52]. In our study, some purines and pyrimidines were downregulated in soybean roots under Mn stress. This finding indicates that purine and pyrimidine metabolisms were inhibited by Mn stress. In addition, linoleic acid and alpha-linolenic acid metabolisms are closely associated with metal. Methyl jasmonate is an intermediate of alpha-linolenic acid metabolism. Studies show that the exogenous application of methyl jasmonate can decrease the degree of oxidative injury in seedlings of rice under Cd and arsenic stress [53,54]. In the present study, most DMs exhibited low accumulation in linolenic acid and alpha-linolenic acid metabolisms in soybean root when the soybean seedlings were treated with 100 µM Mn (Figure 8). This finding indicates that the intermediates of linolenic acid and alpha-linolenic acid metabolisms were inhibited by Mn toxicity. Thus, the tolerance of soybean to Mn poisoning can be improved by exogenous application of the intermediates.

Based on the above results and discussion, a schematic diagram of the metabolic pathways of soybean roots and leaves in response to Mn toxicity was inferred (Figure 9). It was speculated that the response of soybean leaves to Mn was mainly through influencing TCA cycle, flavonoid synthesis, and amino acid metabolism, and leaves mainly responded to Mn toxicity stress by promoting the increase of metabolites in these pathways (Figure 8 and Figure 9). Nevertheless, the roots mainly responded to Mn toxicity by affecting glycolytic pathways, nucleotide metabolism (purine and pyrimidine metabolism), and fatty acid metabolism (linoleic acid and alpha-linoleic acid metabolism), and the metabolites in these pathways were mainly downregulated in response to Mn toxicity stress (Figure 8 and Figure 9). Therefore, leaves and roots of soybean might have different response mechanisms to Mn toxicity stress. However, how exactly these mechanisms differ will continue to be revealed in future studies.

This research showed the influences of Mn on development, activities of enzymes, metal ion absorption, and metabolic activities of soybean roots and leaves. In particular, this research explored the influence of Mn on the pathway of metabolic of soybean leaves and roots. Thus, the key metabolic pathways in soybean roots and leaves responding to Mn toxicity were determined. Hence, this study might provide valuable information for further exploration on the influences of Mn on crop development, activity of enzyme and metabolic activity, and complex regulatory mechanisms presented in soybean roots and leaves suffering Mn poisoning.

## 4. Materials and Methods

### 4.1. Plant Material and Mn Stress Treatment

The soybean YC03-3 (Yuechun03-3) was used in the current study, and the experimental design is shown in Appendix A. First, the seeds were prepared by germination in sand for 4 days. Then, the soybean seedlings were grown in a liquid nutrient solution containing 5 µM (control concentration) and 100 µM (Mn poisoning) MnSO_4_. Then, the seedlings were grown in a greenhouse on a medium with photoperiod of 12 h, at temperatures of 25/20 °C day/night culture for 15 days. The hydroponic nutrient solutions were prepared as described previously [27]. Each experimental treatment consisted of 4 biological replicates. The nutrient solution was updated every 5 days, and the value of pH was regulated to pH = 5.0 every third day. After 15 days of being subjected to Mn stress, the aboveground and underground soybean seedlings were separately harvested. The fresh samples were divided into three parts. The first part was used to analyze the stem width, brown spot measurement, fresh weight, and morphological parameters of the roots. The second part was used to determine the concentrations of Mg, Fe, and Mn in roots and leaves of soybean. The remaining fresh plant materials were quick-frozen with liquid nitrogen and kept in an ultra-low-temperature refrigerator (−80 °C) to measure the activity of antioxidant enzymes and the metabolites of soybean leaves and roots.

### 4.2. Testing of Chlorophyll Contents, Mn Oxide Spots, Metal Ion Contents, and Morphological Root Parameters 

Leaf chlorophyll a and chlorophyll b were determined following the method of the previous study [55]. The collected leaves of the soybean seedlings were weighed and ground into fine powder using a mortar and pestle. The chlorophyll was extracted using 75.0% ethanol with 0.05% (*v*/*v*) Triton X-100 (Beyotime, Shanghai, China) for 24 h in a dark place. The solution absorbance was measured at 665 and 649 nm. Content of chlorophyll was tested on the basis of the below calculation formula [55,56]:Chlorophyll a (μg/mL) = 13.95 × A665 − 6.88 × A649. 
Chlorophyll b (μg/mL) = 24.96 × A649 − 7.32 × A665.

Mn toxicity spots were measured from soybean leaves under different Mn treatments for 15 days. In a nutshell, the brown spots distribution in the upside, middle part, and bottom of leaves were measured by a plastic square with an area of 1 cm^2^, and the average value of Mn spots number from three regions of leaf was calculated. Finally, the final results were calculated from four replicate experiments [14]. The Fe, Mg, and Mn ion concentrations of soybean leaves and roots in each treatment were determined as previously described [57]. Fresh roots and leaves under 5 and 100 µM Mn treatments were separately harvested after 15 days. Then, they were dried and crushed. Approximately 0.2 g of the samples were used to test the ion concentration and the morphological parameters of the soybean via atomic absorption spectrometry. The morphological parameters of the soybean roots were detected with the software of WinRHIZO (2013 e Professional Edition) (Regent Instruments, Quebec, QC, Canada) with a scanner (Epson Expression V800, Epson, CA, USA) [58]. 

### 4.3. Physiological Parameter Measurement

The activities of the antioxidant enzymes (POD, CAT, APX, and SOD) in soybean leaves and roots were tested according to the method described by previous studies [59,60,61,62]. Approximately 0.1 g of fresh roots and leaves were pulverized into fines with a porcelain mortar and liquid nitrogen. The relevant extraction solution (1 mL) was supplemented to the porcelain mortar. The mixture was centrifuged at 10,000× *g* for 10 min at a temperature of 4 °C. The supernatant liquid was applied to determine antioxidant activities. The contents of soluble sugar, malondialdehyde (MDA), soluble protein, and proline were also determined. Proline content was estimated to evaluate the impacts of osmolytes in leaves and roots on the basis of the method by Bates et al. (1973) [63]. The lipid peroxidation level was expressed as the content of MDA via applying TBA (thiobarbituric acid) on the basis of the method by Cakmak and Horst (1991) [64]. The content of soluble protein was determined on the basis of the means by De et al. (2021) [65]. The content of soluble sugar was measured using the method by Fairbairn et al. (1953) [66].

### 4.4. Metabolite Analysis of Soybean Leaves and Root Extracts

The metabolites of soybean leaves and roots were analyzed via LC–MS. The preparation of sample, analysis of LC–MS, and analysis of multivariate are detailed in this section. For the sample preparation, the soybean leaves and roots under 5 and 100 µM Mn stress were used for metabolome analysis. Each treatment was sampled after cultivation for 15 days, and four biological repetitions were employed for each treatment. A proper amount of each sample was weighed precisely before moving it into a centrifuge tube (2 mL cubage). Then, 600 µL MeOH, kept at a temperature of −20 °C, containing 4 ppm 2-Amino-3-(2-chloro-phenyl)-propionic acid, was supplemented and vortex-oscillated for 30 s. Moreover, 100 mg glass beads were supplemented and moved into a tissue grinder at 60 Hz for 90 s. The ultrasonic treatment (at 25 °C) was performed for 15 min, followed by centrifuging treatment at 4 °C for 10 min at the speed of 12,000 rpm. The supernatant liquid was filtered by a 0.22 µm filtering membrane and moved into the test bottle for LC–MS (liquid chromatography–mass spectrometry) testing. The LC judgment was implemented on a system of ultra-high-performance liquid chromatography (UHPLC) (Thermo Fisher Scientific, USA). Mass spectrometric testing of metabolic products was implemented on Q Exactive (Thermo FisherScientific, USA) with an ESI ion source.

For the analyses of LC–MS and multivariate, the original data were firstly transformed into mzXML format via the MS Convert in ProteoWizard software package (v3.0.8789) [67]. They were further processed with XCMS (various forms (X) of chromatography mass spectrometry) [68] for property detection, keeping time correction, and alignment. The metabolic products were authenticated by accuracy mass (<30 ppm), and the data of MS/MS were further matched with the HMDB (human metabolome database) (http://www.hmdb.ca, accessed on 27 August 2023) [69], the massbank (http://www.massbank.jp/, accessed on 27 August 2023) [70], the lipidMaps (http://www.lipidmaps.org, accessed on 27 August 2023) [71], the mzclou (https://www.mzcloud.org, accessed on 27 August 2023) [72], and the KEGG (http://www.genome.jp/kegg/, accessed on 27 August 2023) [73]. The robust LOESS signal correction (QC-RLSC) [74] was used for normalization of data to remedy systematic bias. After normalization processing, only peaks of ion with relativity SD (standard deviations) of below 30% in quality control (QC) were reserved to guarantee the correct identification of metabolic products.

Ropls software (version 3.0.2) was applied to all analyses of multivariate data and models [75]. After scaling data, the models were built based on the PCA (principal component analysis), the PLS-DA (partial least-square discriminant analysis), and the OPLS-DA (orthogonal partial least-square discriminant analysis). The metabolic profiles were visualized as score plots, in which each point represented a sample. The relevant loading plot and S-plot were created to offer information on the metabolic products that affected sample clustering. All of the models evaluated were verified for overfitting by permutation inspection. The descriptive performance of the models was tested by R2X (cumulative) (perfect model: R2X (cum) = 1) and R2Y (cumulative) (perfect model: R2Y (cum) = 1) values. Their prediction performance was tested by Q2 (cumulative) (perfect model: Q2 (cum) = 1) and a permutation inspection. The permuted model might not be capable of predicting classes. The R2 and Q2 values at the *Y*-axis intercept should be lower than those of Q2 and the R2 of the model of nonpermutation. OPLS-DA permitted the testing of the discriminating metabolites by the VIP. The *p* value, VIP produced by OPLS-DA, and FC were adopted to find the contributable variable for classification. In the end, *p* value < 0.05 and VIP values > 1 were recognized as metabolic products with statistical significance. DMs were applied to analysis of pathway with the help of MetaboAnalyst [76], which integrated the findings from strong analysis of pathway enrichment with the analysis of pathway topology. Then, the identified metabolic products in metabolomics were reflected in the pathway of KEGG (Kyoto Encyclopedia of Genes and Genomes) for biological analysis of advanced systemic functions. The metabolic products and relevant pathways were directly visualized by the KEGG Mapper tool.

### 4.5. Analysis of Data

All of the tests in the current study were performed with four biological duplications, and the analysis of data was adopted with the software of SPSS (version 19.0, SPSS Inc., Chicago, IL, USA). All data were adopted with a one-way analysis of variance (Student’s *t*-test). Data were represented with the average value ± SD (standard deviation); * *p* ≤ 0.05, ** *p* ≤ 0.01.

## 5. Conclusions

This research shows that Mn toxicity markedly decreased the development and fresh weight of soybean in aboveground and underground tissues. In addition, Mn stress altered the activities of antioxidant enzymes (APX, POD, CAT, and SOD) and the contents of metallic ions (Fe, Mg, and Mn) in soybean leaves and roots. Mn also changed the metabolite levels of soybean roots and leaves by adjusting and controlling the metabolic pathways. In total, 136 and 164 metabolites were screened for changes in soybean roots and leaves under Mn stress; however, the kinds of metabolites were markedly different. In soybean roots, the metabolisms of linoleic acid, purine, glycolysis/gluconeogenesis, alpha-linoleic acid, and pyrimidine were mainly involved; in soybean leaves, the metabolites changed in ABC transporters, citrate cycle (TCA cycle), oxidative phosphorylation, flavonoid biosynthesis, and metabolisms of glutamate, aspartate, and alanine. Nevertheless, the specific mechanisms relevant to those effective results must be verified by further in-depth study.

## Figures and Tables

**Figure 1 plants-12-03615-f001:**
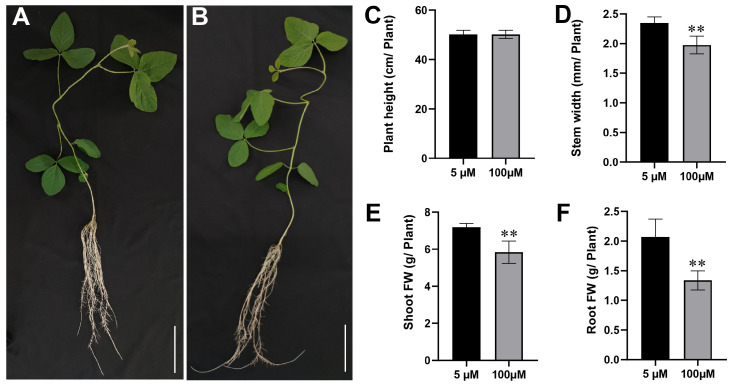
Effects of soybean growth with Mn treatment. Morphologies of soybean plants at various concentrations of Mn: (**A**) 5 μM; (**B**) 100 μM (bars = 5 cm). (**C**) Soybean plant height; (**D**) soybean plant stem width; (**E**) fresh weight of shoot; (**F**) fresh weight of root (** *p* ≤ 0.01).

**Figure 2 plants-12-03615-f002:**
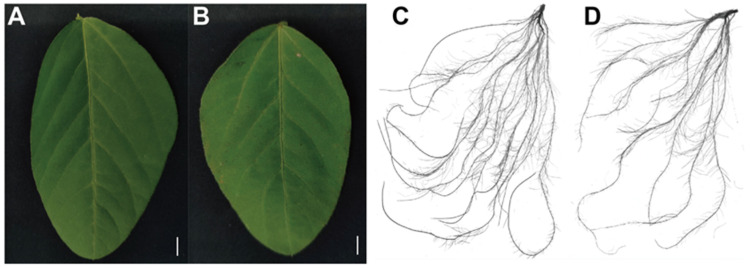
Phenotypes of soybean leaves and roots at different Mn concentrations: (**A**) leaf phenotypes (5 μM); (**B**) leaf phenotypes (100 μM); (**C**) root phenotypes (5 μM); (**D**) root phenotypes (100 μM). Bars = 2 cm.

**Figure 3 plants-12-03615-f003:**
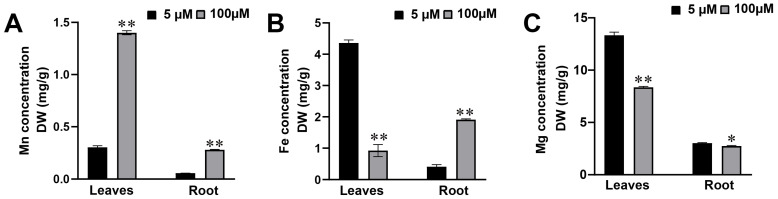
Contents of Mg, Fe, and Mn in roots and leaves of soybean with various Mn treatment. Contents of (**A**) Mn; (**B**) Fe; (**C**) Mg (* *p* ≤ 0.05, ** *p* ≤ 0.01).

**Figure 4 plants-12-03615-f004:**
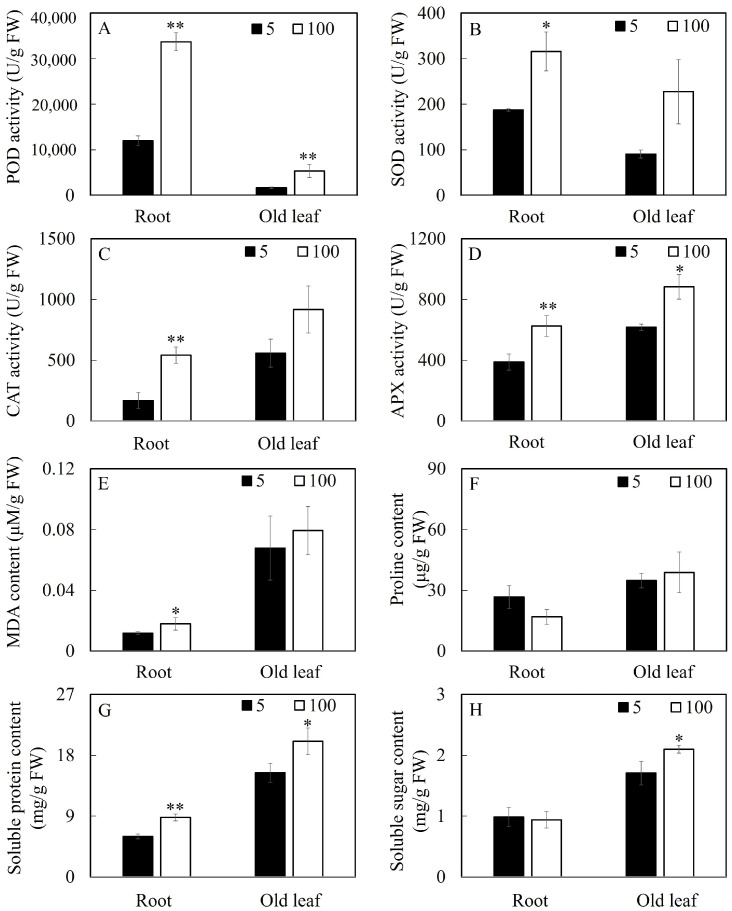
Results of various Mn levels on physiological parameters in soybean roots and leaves. Activity of (**A**) POD, (**B**) SOD, (**C**) CAT, and (**D**) APX; (**E**) content of MDA, (**F**) Pro, (**G**) soluble protein, and (**H**) soluble sugar. (* *p* ≤ 0.05, ** *p* ≤ 0.01).

**Figure 5 plants-12-03615-f005:**
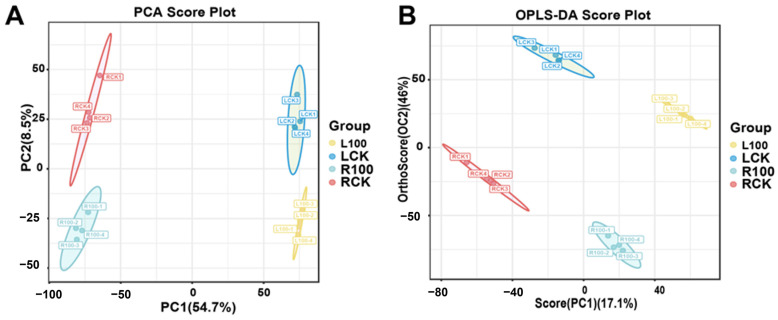
The statistical analysis of soybean leaves and roots metabolome data at different Mn treatment concentrations. (**A**) PCA; (**B**) OPLS-DA. LCK, L100, RCK, and R100 represent 5 µM Mn leaves, 100 µM Mn leaves, 5 µM Mn roots, and 100 µM Mn roots, respectively.

**Figure 6 plants-12-03615-f006:**
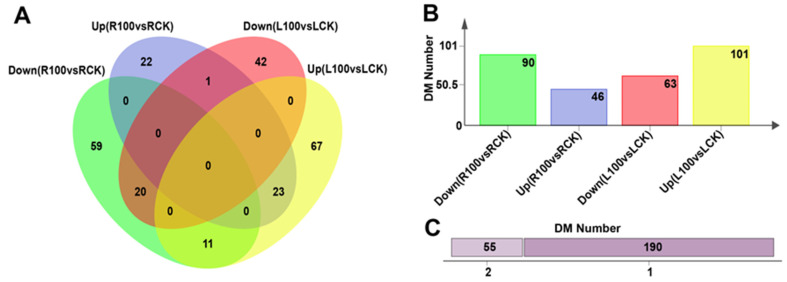
The statistical analysis of DM data in leaves and roots at different Mn treatment concentrations. (**A**) Venn diagram showing the distribution and relationship of DMs between groups; (**B**) the number of DM in each group; (**C**) number of DMs: specific (1) or shared (2).

**Figure 7 plants-12-03615-f007:**
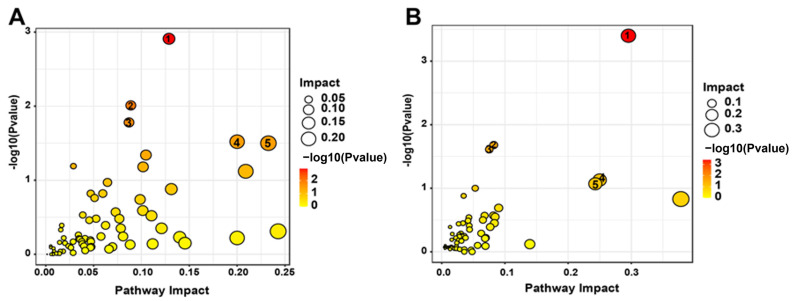
Bubble diagrams of metabolism pathways of DMs in soybean leaves and roots. (**A**) L100 vs. LCK; (**B**) R100 vs. RCK. Each bubble represents a kind of metabolism pathway. The darker and larger the bubble is, the stronger the association between the metabolic pathway and Mn stress is. Larger size and darker color of the bubbles represent a stronger correlation between the metabolism pathway and Mn poisoning.

**Figure 8 plants-12-03615-f008:**
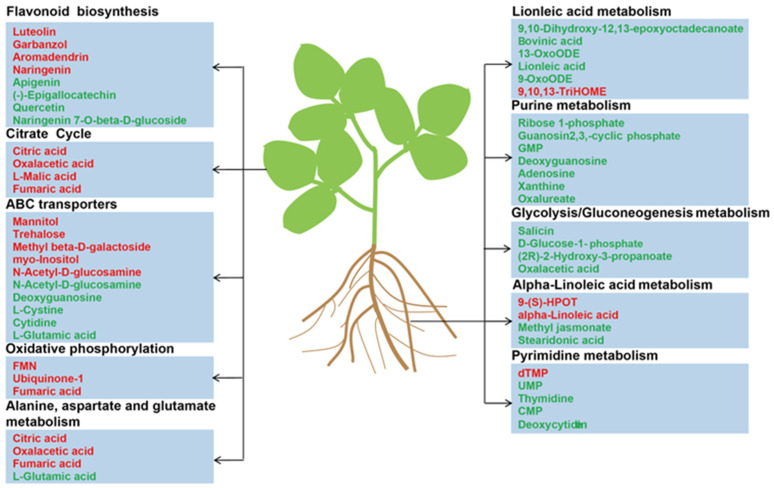
The major metabolites in soybean leaves and roots responding to Mn poisoning. The red represents the increased metabolites, whereas the green represents the decreased metabolites.

**Figure 9 plants-12-03615-f009:**
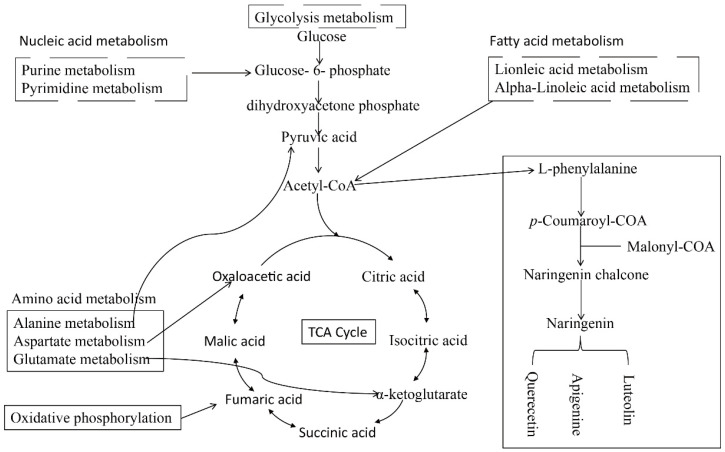
Schematic diagram of metabolic pathways of soybean root and leaf in response to Mn toxicity. The dashed line frame represents the metabolic pathway of the root in response to Mn toxicity, and the solid line frame represents the metabolic pathway of the leaf in response to Mn toxicity.

**Table 1 plants-12-03615-t001:** Influences of different Mn concentrations on the roots and leaves of soybean.

Parameters of Soybean Root Growth	Concentrations of Mn (μM)	
	5	100
Chlorophyll a content	2.48 ± 0.01	2.47 ± 0.038
Chlorophyll b content	1.72 ± 0.13	1.70 ± 0.15
Number of manganese spots on the old leaves	0	31.75 ± 1.21 **
Average diameter of root (mm)	0.93 ± 0.005	0.86 ± 0.034 *
Volume of root (cm^3^)	37.19 ± 2.81	23.38 ± 3.70 **
Surface area of root (cm^2^)	1104.10 ± 85	694.64 ± 82.07 **
Total length of root (cm)	4220.04 ± 359.85	2821.86 ± 220.87 **
Root tip number	3204.75 ± 256.38	3889.25 ± 814.38

Notes: Data were represented by the average value and standard deviation of four times experimental replications. Student’s *t*-test was used to assess the significance of the difference between the control and Mn toxicity (* *p* ≤ 0.05, ** *p* ≤ 0.01).

## Data Availability

The raw metabolomics data were deposited in the MetaboLights (https://www.ebi.ac.uk/metabolights/, accessed on 27 August 2023) under accession number MTBLS8472.

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
