# Peer review of "Metabolic Responses to Manganese Toxicity in Soybean Roots and Leaves"

_plants, 2023, doi:10.3390/plants12203615_

Round 1

Reviewer 1 Report

Mn is essential to plants for growth and development. However, at higher concentrations Mn becomes toxic. As soybeans are important agricultural produce, it is important to find the Mn levels, which become detrimental.

The present paper is very informative, as it also describes research on toxicity mechanisms, as Mn concentration rises.

The paper is well written, but some minor English editing might be needed.

For instance: "Mn is one kind of trace elements essential to plants"

Author Response

Reviewer 1

Mn is essential to plants for growth and development. However, at higher concentrations Mn becomes toxic. As soybeans are important agricultural produce, it is important to find the Mn levels, which become detrimental.

The present paper is very informative, as it also describes research on toxicity mechanisms, as Mn concentration rises.

The paper is well written, but some minor English editing might be needed.

For instance: "Mn is one kind of trace elements essential to plants"

My response: That's good advice, thank you.

Yes! I have carefully revised the English editing of the whole article. The changes have been marked in red or blue or purple.

Reviewer 2 Report

Soybean is an enormously important crop that is widely cultivated throughout the world. Research on the mechanisms of stress tolerance in soybean is essential for improving productivity in the future and is being investigated from various aspects. Heavy metal stress in soil is known to play a significant factor in soybean productivity, and many researchers are interested in the response of soybean to crucial heavy metal elements, their dynamic behaviour in the body and the stress tolerance mechanisms.

This paper studies the response of soybean leaves and roots to excess Mn. The authors performed metabolomic analysis in addition to growth parameters, heavy metal content and enzyme activities involved in typical antioxidants. The metabolomic analysis revealed that excess Mn affects differential primary metabolic pathways in the leaves and roots of soybean, respectively.

Although the results of this work are fascinating, it is still considered to be at a preliminary phase before being published as an article in this journal. The following are the major points.

1) The results of the metabolome analysis show that excess Mn exposure affects different primary metabolic pathways in leaves and roots. What does this mean? Are different metabolic pathways impaired in leaves and roots, or is this the result of different Mn tolerance mechanisms expressed in leaves and roots? It is essential to present the authors' discussion to help the reader understand the implications of this event.

(2) The authors have shown that excess Mn exposure affects different primary metabolic pathways in leaves and roots. However, it needs to be clarified whether this is Mn-specific or common in other heavy metals. This is important for this exciting study to consider and would be of interest to many researchers.

Author Response

Reviewer 2

Soybean is an enormously important crop that is widely cultivated throughout the world. Research on the mechanisms of stress tolerance in soybean is essential for improving productivity in the future and is being investigated from various aspects. Heavy metal stress in soil is known to play a significant factor in soybean productivity, and many researchers are interested in the response of soybean to crucial heavy metal elements, their dynamic behaviour in the body and the stress tolerance mechanisms.

This paper studies the response of soybean leaves and roots to excess Mn. The authors performed metabolomic analysis in addition to growth parameters, heavy metal content and enzyme activities involved in typical antioxidants. The metabolomic analysis revealed that excess Mn affects differential primary metabolic pathways in the leaves and roots of soybean, respectively.

Although the results of this work are fascinating, it is still considered to be at a preliminary phase before being published as an article in this journal. The following are the major points.

1) The results of the metabolome analysis show that excess Mn exposure affects different primary metabolic pathways in leaves and roots. What does this mean? Are different metabolic pathways impaired in leaves and roots, or is this the result of different Mn tolerance mechanisms expressed in leaves and roots? It is essential to present the authors' discussion to help the reader understand the implications of this event.

My response: The result 2.7 means is different Mn tolerance mechanisms expressed in leaves and roots, and Figure 8 and 9 illustrates this result. Besides, this result is discussed in Line 315-331.

(2) The authors have shown that excess Mn exposure affects different primary metabolic pathways in leaves and roots. However, it needs to be clarified whether this is Mn-specific or common in other heavy metals. This is important for this exciting study to consider and would be of interest to many researchers.

My response: Thank you for your good advice. But, this paper mainly focuses on whether the response of soybean roots and leaves to Mn toxicity is the same. Compared to other heavy metals, common metabolic pathways were discussed in Line 320-331, and Mn specific metabolic pathways will be studied in future studies.

Reviewer 3 Report

The ms entitled Metabolic responses to manganese toxicity in soybean roots and leaves investigates an important topic but the authors should revise it before it can be accepted in such high quality journal:

General notes: The authors have to follow the guidelines of the authors to make the life easier for review. Now, there is no line numbers, and was difficult for me to address specific points and comments as much as I can.

I really was looking to find something that could be used to mitigate Mn stress in the current study, but I did not find.

Abstract:

1-     Use “Soybean is one of …” instead of “Soybeans are

2-     Why authors decided to work on manganese? Because soil does not have high level of available manganese, this means that this element is not critical issue for plants or environment.

3-     Why authors use 5 and 100 μM of Mn? Why not 50 and 100 μM of Mn? Why now 25, 50 and 100 μM of Mn?

4-     the antioxidant enzyme activities and the soluble protein content in leaves and roots of soybean were all increased!! Which enzymes exactly? Please specify.

5-     Why authors analyzed Fe and Mg and did not analyze the Zn or B or other microelements?

Keywords

6-     Use the Latin name instead of Soybean

Introduction

7-    Go through the introduction and leave space between last word in the sentence and the [ such as plants[1].

8-    30 to 500 mg per kilogram, use kg instead of kilogram.

9-    Plants produce various physiological changes and biochemical responses to relieve the poisoning symptoms of metal ion excess. Add this citation at end of this sentence “Enhancing antioxidant defense system of mung bean with a salicylic acid exogenous application to mitigate cadmium toxicity”

10- Zhang et al. (2018) conducts metabolomics analysis, use the correct format to cite Zhang et al.

11- What do you mean by “and so on”? in “citraconic acid, acetanilide, lactamide, raffinose, and so on”

12- Soybean is an important oil crop and a source of animal feed[24]. Please add the Latin name of soybean and also cite this ref. “Exogenous potassium treatments elevate salt tolerance and performances of Glycine max L. by boosting antioxidant defense system under actual saline field conditions”

13- Revise the aim at the end of the introduction section.

Results

14- Remove “Two Mn concentrations, 5 μM (normal Mn) and 100 μM (Mn poisoning), were adopted to investigate the influences of Mn poisoning on soybean growth.” We already know that.

15- Fig 1 E, Please revise the title of the axis to  Shoot FW (g / plant). Also, change the title of axis in Fig 1 F to Root FW (g / plant)

16-  What do you refer for (bars = 5 cm) in the title of Figure 1?

17- Figure 3, The authors should add at the end of the axis title FW or DW?

18- In title of Figure 3: authors should use (* p 0.05, ** p 0.01) instead of (* p < 0.05, ** p < 0.01)

19- See previous comments to revise title of Figure 4.

Discussion

20- I need the authors to focus deeply on the mechanisms of the Mn toxicity and how they negatively affected the plant biochemical and physiological traits and link this with their own results.

Material and methods

21- Based one the period of the experiment as authors said “After 15 days of being subjected to Mn stress, ….” I can not imagine how plants can give FW of shoots or roots as shown in the Figures within the results section. I really can not imagine. I hope the authors can check this issue.

22- Why authors decided to work on these two levels of Mn? Because the lowest level is very small and these big range between both of concentration.

23- Please revise the design of the experiment, can you insert a photo for your experiment in the greenhouse here?

Conclusion

24- In this section, the authors should straight focus on what they found. Therefore, remove the following text “This research provided available information on the influences of Mn toxicity on soybean development, morphology of root, response of antioxidant, metal ion content, and metabolisms of soybean roots and leaves.”

25- I also, recommend the authors to add some key data in the conclusion to highlight their findings.

Good luck

minor editing is needed

Author Response

Reviewer 3

The ms entitled Metabolic responses to manganese toxicity in soybean roots and leaves investigates an important topic but the authors should revise it before it can be accepted in such high quality journal: General notes: The authors have to follow the guidelines of the authors to make the life easier for review. Now, there is no line numbers, and was difficult for me to address specific points and comments as much as I can. I really was looking to find something that could be used to mitigate Mn stress in the current study, but I did not find.

My response: Thank you for your suggestion. I have added the line number in the revision manuscript.

Abstract:

1-     Use “Soybean is one of …” instead of “Soybeans are”

My response: Line 8. OK, I've already changed Soybeans are” to “Soybean is one of …”

2-     Why authors decided to work on manganese? Because soil does not have high level of available manganese, this means that this element is not critical issue for plants or environment.

My response: Mn is also considered one of the heavy metals that can be harmful to plants at excessive levels. When the Mn concentration in the aboveground tissues of plants reaches 150 mg per kilogram dry weight, Mn toxicity can generally occur, especially for plants growing in acid soils. However, the concentration of Mn varies between 40 and 1681 mg per kilogram in farmland soils across mainland China according to reference “Advances in the Mechanisms of Plant Tolerance to Manganese Toxicity” (Li et al., 2019) (Line 492-493), so we decided to work on manganese.

3-     Why authors use 5 and 100 μM of Mn? Why not 50 and 100 μM of Mn? Why now 25, 50 and 100 μM of Mn?

My response: According to previous study (Liu et al., 2020) (Line 549-550). Brown spots, root fresh weight and shoot fresh weight changed significantly at 100 µM Mn concentration in soybean (as manganese poisoning test group), while 5 µM is the manganese concentration suitable for the normal growth of soybeans (as control group). However, when the concentration of manganese treatment is 25-50 µM, soybeans will also suffer different degrees of manganese toxicity, and there will be obvious manganese oxidation spots on the leaves. So finally, our ultimately chose 5 µM (control group) and 100 µM (manganese poisoning test group) Mn concentration.

4-     the antioxidant enzyme activities and the soluble protein content in leaves and roots of soybean were all increased!! Which enzymes exactly? Please specify.

My response: I have described antioxidant enzymes in detail (Line 14-15).

“the activity of superoxide dismutase (SOD), catalase (CAT), peroxidase (POD), ascorbate peroxidase (APX)”

5-     Why authors analyzed Fe and Mg and did not analyze the Zn or B or other microelements?

My response: According to the reference 9-11 (Line 508-513), Mn stress can inhibit the absorption and transport of other elements iron [Fe] and magnesium [Mg]. (Line 43-47)

Keywords

6-     Use the Latin name instead of Soybean

My response: OK. I have used the Latin name instead of Soybean. (Line 27)

Introduction

7-    Go through the introduction and leave space between last word in the sentence and the [ such as plants[1].

My response: Thank you for your suggestion. Questions like this have been revised in the entire text.

8-    30 to 500 mg per kilogram, use kg instead of kilogram.

My response: I have changed kilogram to kg (Line 41 and 43). The changes have been marked in purple and blue.

9-    Plants produce various physiological changes and biochemical responses to relieve the poisoning symptoms of metal ion excess. Add this citation at end of this sentence “Enhancing antioxidant defense system of mung bean with a salicylic acid exogenous application to mitigate cadmium toxicity”

My response: I have added this reference (Line 52, Line 523-525).

10- Zhang et al. (2018) conducts metabolomics analysis, use the correct format to cite Zhang et al.

My response: Line 67-69. I have modified it as suggested.

“Zhang et al. conducts metabolomics analysis to reveal how cucumbers (Cucumis sativus) reprogram metabolic products to handle silver (Ag+) and silver nanoparticles (AgNPs) that induce oxidative stress (2018).”

11- What do you mean by “and so on”? in “citraconic acid, acetanilide, lactamide, raffinose, and so on”

My response: The sentence in line 69-72, “and so on” means that there are many metabolites induced by AgNPs, some of which are not listed

12- Soybean is an important oil crop and a source of animal feed[24]. Please add the Latin name of soybean and also cite this ref. “Exogenous potassium treatments elevate salt tolerance and performances of Glycine max L. by boosting antioxidant defense system under actual saline field conditions”

My response: I have added the Latin name of soybean and cite the reference (Line 76, Line 546-548), The changes have been marked in red.

13- Revise the aim at the end of the introduction section.

My response: L85-94. I have amended the paragraph as suggested.

However, metabolomics analysis of soybean in response to Mn poisoning is very few, while metabolome analysis of soybean leaves and roots exposed to manganese toxicity has not been reported. In current study, the phenotypic changes and physiological response indexes of soybean leaves and roots after high and normal Mn treatment were analyzed. Then, in order to further study the specific metabolic regulation mechanism of soybean subjected to manganese toxicity stress, the changes in metabolites with normal and high concentrations of Mn treatment were analyzed through metabolomics analysis.

Results

14- Remove “Two Mn concentrations, 5 μM (normal Mn) and 100 μM (Mn poisoning), were adopted to investigate the influences of Mn poisoning on soybean growth.” We already know that.

My response: Thank you, I have deleted that sentence (Line 99-100).

15- Fig 1 E, Please revise the title of the axis to Shoot FW (g / plant). Also, change the title of axis in Fig 1 F to Root FW (g / plant)

My response: Thank you for your advice. Line 106-108.

I have changed the title of axis in Fig 1 E and Fig 1 F to Shoot FW (g / plant) and Root FW (g / plant).

16-  What do you refer for (bars = 5 cm) in the title of Figure 1?

My response: bars = 5 cm means that the scale (white line) in Figures 1A and B is 5 cm.

17- Figure 3, The authors should add at the end of the axis title FW or DW?

My response: Line 141-142. I have added at the end of the axis title DW in Figure 3.

18- In title of Figure 3: authors should use (* p ≤0.05, ** p ≤0.01) instead of (* p < 0.05, ** p < 0.01)

My response: Thank you, I have use (* p ≤0.05, ** p ≤0.01) instead of (* p < 0.05, ** p < 0.01). The changes have been marked in red (Line 111, 129, 144, 151, 161, 452)

19- See previous comments to revise title of Figure 4.

My response: I have use (* p ≤0.05, ** p ≤0.01) instead of (* p < 0.05, ** p < 0.01). The changes have been marked in red (Line 160-161).

Discussion

20- I need the authors to focus deeply on the mechanisms of the Mn toxicity and how they negatively affected the plant biochemical and physiological traits and link this with their own results.

My response: This is a good suggestion. Schematic diagram of metabolic pathways of soybean root and leaf in response to Mn toxicity have been shown as follows. (Line 315-331)

Figure 9 Schematic diagram of metabolic pathways of soybean root and leaf in response to Mn toxicity.

The dashed line frame represents the metabolic pathway of the root in response to Mn toxicity, and the solid line frame represents the metabolic pathway of the leaf in response to Mn toxicity.

Based on the above results and discussions, a schematic diagram of the metabolic pathways of soybean roots and leaves in response to Mn toxicity was inferred (Figure 9). It was speculated that the response of soybean leaves to Mn was mainly through influencing TCA cycle, flavonoid synthesis and amino acid metabolism, and leaves were mainly responded to Mn toxicity stress by up-regulation metabolite in these pathways (Figure 8 and 9). Nevertheless, the roots were mainly responded to Mn toxicity by affecting glycolytic pathways, nucleotide metabolism (purine and pyrimidine metabolism), and fatty acid metabolism (lionleic acid and alpha-linoleic acid metabolism), and the metabolites in these pathways were mainly down-regulated in response to Mn toxicity stress (Figure 8 and 9). Therefore, leaves and roots of soybean might have different response mechanisms to Mn toxicity stress. However, how exactly these mechanisms differ will continue to be revealed in future studies.

Material and methods

21- Based one the period of the experiment as authors said “After 15 days of being subjected to Mn stress, ….” I can not imagine how plants can give FW of shoots or roots as shown in the Figures within the results section. I really can not imagine. I hope the authors can check this issue.

My response: Line 343-344. The experimental design was shown in Figure S4.

22- Why authors decided to work on these two levels of Mn? Because the lowest level is very small and these big range between both of concentration.

My response: According to previous study (Liu et al., 2020) (Line 549-550). Brown spots, root fresh weight and shoot fresh weight changed significantly at 100 µM Mn concentration in soybean (as manganese poisoning test group), while 5 µM is the manganese concentration suitable for the normal growth of soybeans (as control group). However, when the concentration of manganese treatment is 25-50 µM, soybeans will also suffer different degrees of manganese toxicity, and there will be obvious manganese oxidation spots on the leaves. So finally, our ultimately chose 5 µM (control group) and 100 µM (manganese poisoning test group) Mn concentration.

23- Please revise the design of the experiment, can you insert a photo for your experiment in the greenhouse here?

My response: OK. Detailed experimental design has been shown in FIG. S4. Line 343-344.

A photo for our experiment in the greenhouse has been shown as follow:

Conclusion

24- In this section, the authors should straight focus on what they found. Therefore, remove the following text “This research provided available information on the influences of Mn toxicity on soybean development, morphology of root, response of antioxidant, metal ion content, and metabolisms of soybean roots and leaves.”

My response: Thank you, I have removed the sentence (Line 455-457).

25- I also, recommend the authors to add some key data in the conclusion to highlight their findings.

My response: Yes, we have added some key data in conclusion (Line 315-331).

Reviewer 4 Report

In this manuscript, a hydroponic experiment was conducted on the growth, development, enzyme activity, and metabolism of soybeans under varying levels of Mn treatment (5 and 100 μM). Compared with the control, the soybeans under Mn stress showed inhibited growth and development.

Moreover, the activities of antioxidases and the content of soluble protein in roots and leaves were increased. However, the content of soluble sugar and proline in roots and leaves showed the opposite trend. In addition, the iron (Fe) and magnesium (Mg) ion contents in leaves significantly decreased, and the Mn ion content greatly increased. The metabolomic analysis based on nontargeted liquid chromatography–mass spectrometry identified 136 and 164 differential metabolites (DMs) responding to Mn toxicity in soybean roots and leaves. These DMs were involved in five different main metabolic pathways in soybean roots and leaves, indicating that soybean roots and leaves exhibited various toxicity responses to Mn toxicity. These findings indicate that Mn toxicity changes the activity of enzymes and affects the metabolic pathway, thereby inhibiting the growth of soybeans and leading to toxicity.

- The study has been well performed and provided some important and useful results. However, some revisions are needed as revealed below;

- There are many mistakes in the English language in the manuscript. The manuscript must be revised and corrected by either English editing service or English native speaker.

- The methods should be written in more detail to be reproducible. How many biological and technical replications used?

- The discussion should be improved and interpreted with the results and discussed in relation to the current literature.

- The references section should include update references

Extensive editing of English language required

Author Response

Reviewer 4

In this manuscript, a hydroponic experiment was conducted on the growth, development, enzyme activity, and metabolism of soybeans under varying levels of Mn treatment (5 and 100 μM). Compared with the control, the soybeans under Mn stress showed inhibited growth and development.

Moreover, the activities of antioxidases and the content of soluble protein in roots and leaves were increased. However, the content of soluble sugar and proline in roots and leaves showed the opposite trend. In addition, the iron (Fe) and magnesium (Mg) ion contents in leaves significantly decreased, and the Mn ion content greatly increased. The metabolomic analysis based on nontargeted liquid chromatography–mass spectrometry identified 136 and 164 differential metabolites (DMs) responding to Mn toxicity in soybean roots and leaves. These DMs were involved in five different main metabolic pathways in soybean roots and leaves, indicating that soybean roots and leaves exhibited various toxicity responses to Mn toxicity. These findings indicate that Mn toxicity changes the activity of enzymes and affects the metabolic pathway, thereby inhibiting the growth of soybeans and leading to toxicity.

- The study has been well performed and provided some important and useful results. However, some revisions are needed as revealed below;

- There are many mistakes in the English language in the manuscript. The manuscript must be revised and corrected by either English editing service or English native speaker.

My response: Thank you for your helpful advice. We have asked professionals to help us polish the English of our manuscript.

- The methods should be written in more detail to be reproducible. How many biological and technical replications used?

My response: This study used four biological replicates (Line 349-350)

- The discussion should be improved and interpreted with the results and discussed in relation to the current literature.

My response: This issue has already been discussed in Line 315-331.

- The references section should include update references

My response: The references already include 25 articles from 2020 and 2023 (Line 492, 496, 520, 530, 536, 542, 544, 546, 549, 553, 565, 567, 569, 573, 584, 586, 589, 596, 602, 606, 619, 621, 627, 642)

Round 2

Reviewer 2 Report

The authors of this paper responded politely to the reviewers' points. The text was made easier to read and important aspects of the research were highlighted. The paper will provide important information for researchers studying heavy metal tolerance in plants.

Reviewer 4 Report

The revised manuscript has been improved as per suggested revisions